# Functional Activity in the Effect of Transcranial Magnetic Stimulation Therapy for Patients with Depression: A Meta-Analysis

**DOI:** 10.3390/jpm13030405

**Published:** 2023-02-24

**Authors:** Yongyan Deng, Wenyue Li, Bin Zhang

**Affiliations:** 1The Affiliated Brain Hospital of Guangzhou Medical University, Guangzhou 510370, China; 2Peking University Sixth Hospital, Beijing 100191, China; 3Institute of Mental Health, Tianjin Anding Hospital, Tianjin Medical University, Tianjin 300222, China

**Keywords:** transcranial magnetic stimulation, depression, neuroimaging, activation likelihood estimation, systematic review

## Abstract

Depression is a long-lasting mental disorder that affects more than 264 million people worldwide. Transcranial magnetic stimulation (TMS) can be a safe and effective choice for the treatment of depression. Functional neuroimaging provides unique insights into the neuropsychiatric effects of antidepressant TMS. In this meta-analysis, we aimed to assess the functional activity of brain regions caused by TMS for depression. A literature search was conducted from inception to 5 January 2022. Studies were then selected according to predetermined inclusion and exclusion criteria. Activation likelihood estimation was applied to analyze functional activation. Five articles were ultimately included after selection. The main analysis results indicated that TMS treatment for depression can alter the activity in the right precentral gyrus, right posterior cingulate, left inferior frontal gyrus and left middle frontal gyrus. In resting-state studies, increased activation was shown in the right precentral gyrus, right posterior cingulate, left inferior frontal gyrus and left superior frontal gyrus associated with TMS treatment. In task-related studies, clusters in the right middle frontal gyrus, left sub-gyrus, left middle frontal gyrus and left posterior cingulate were hyperactivated post-treatment. Our study offers an overview of brain activity changes in patients with depression after TMS treatment.

## 1. Introduction

Depression is a common mental disorder that affects more than 264 million people worldwide and is characterized by a constant state of sadness and loss of interest or pleasure in previously rewarding or enjoyable activities [1]. It is a long-lasting and recurrent mental disorder that can affect people in terms of both physical and mental dysfunction, presenting as cognitive impairment and long-term memory dysfunction, which strongly predict work disability and premature death [2,3,4,5]. Depression increases the risk of premature mortality due to cardiovascular diseases and metabolic, psychiatric and addictive comorbidities that are associated with a worse prognosis [6,7,8].

Treatments for depression have long been a complicated problem, even with years of development. Although there are multiple antidepressant therapies for clinical selection, nearly one-third of patients are non-responders to any existing pharmacotherapy or psychotherapy treatment and remain symptomatically and functionally impaired. In addition, sequential and empirical attempts to identify initial antidepressant treatment resistance can lead to progressive complexity of clinical medication therapy, which may be associated with a diminishing likelihood of remission with each failed treatment attempt [9,10].

Impaired brain plasticity can be the mechanism for depressive mood and other related symptoms, manifesting as anomalous cortical activity and dysregulation of connectivity [11]. The electroencephalogram (EEG) can be an accurate proxy of long-term clinical outcomes for patients with neuropsychiatric disorder that reflects damage to the neural network [12]. Patients with major depression disorder present rhythmic aberrations in prefrontal cortex recorded by EEG, which may be associated with changes occurring in cortical and subcortical loops related to cognition impairment, showing decrease in alpha connectivity especially [13]. This not only relates to depression symptoms of attentional impairment, suicidal ideation, rumination, and depression behavioral scale scores, depression severity and the curative effect of depression treatment, but has also been shown to significantly affect higher cognitive abilities [13]. Brain imaging has also revealed abnormalities in regional brain functioning associated with depression. Functional magnetic resonance imaging (fMRI) is able to detect abnormalities in brain activity, including abnormal neural activity in the anterior cingulate, prefrontal cortex and orbitomeatal cortex in depression associated with cognition impairment [14].

Transcranial magnetic stimulation (TMS), a non-invasive method of stimulating relevant brain regions beneath an applied coil by a magnetic field inducing electrical currents, can be a safe and effective choice for depression treatment [15,16]. TMS can alter brain region cortical activity and propagate its effect by means of existing neural networks [17]. It stimulates an increase in alpha connectivity in depression patients’ EEG implying adjustment of impaired neural functional connectivity, which is associated with symptom improvement [18]. Repetitive TMS (rTMS), a common type of TMS treatment used clinically, can provide long-lasting therapeutic benefit against depressive mood and depression-relevant impairment associated with its capability to modulate cortical excitability and connectivity of the areas underlying mood and affect regulation, which is caused by plastic remodeling of synaptic connections [11,19]. Furthermore, rTMS can normalize intracortical facilitation in patients with major depression by performing single and paired TMS before and after treatment for evaluation, which can result in increased cortical excitability and neurochemical transmission [20]. However, the response to TMS treatment is different for patients with depression and healthy subjects due to patients’ dysfunctional neuroplasticity [21]. Thus, further understanding of the neuropsychiatric effect of antidepression TMS treatment, particularly for patients with depression, is important for the optimal use of TMS. Functional neuroimaging provides unique insights for investigating the neuropsychiatric changes caused by TMS treatment [22].

In recent years, the effect of TMS therapy on brain functional activity changes in depression has been a topic of concern. One study showed that the right orbitofrontal cortex and hippocampus, which are the elements of brain networks thought to be related to depression, became significantly more active in the course of TMS for depression [23]. Another study found that functional activity in the subgenual cingulate cortex attenuated to lower levels with successful TMS treatment in patients with depression [24].

However, the results of these studies are inconsistent. There has been no systematic review or meta-analysis that summarizes the effect of antidepressant TMS treatment on brain functional changes. A quantitative analysis known as activation likelihood estimation (ALE) can synthesize functional activity changes before and after receiving antidepressant TMS treatment. ALE is a coordinator-based meta-analysis method that changes the analysis focus from consistency between foci to agreement across experiments using a random effects algorithm. Instead of treating foci as points, reported coordinators are recognized as spatial probability distribution centers in ALE—computing the union of these centers can produce an ALE map [25]. A permutation test is applied to indicate reliability by obtaining ALE maps in which the null distribution of the same number of foci as in the real analysis is randomly redistributed throughout the brain. The contribution of each study is weighted by its sample size [26,27].

This review aims to give a current overview of the existing literature that explores the particular neuropsychiatric effect of TMS treatment on patients with depression by quantitatively evaluating brain functional activity. In the present study, ALE analysis was conducted to assess the functional activity of brain regions.

## 2. Materials and Methods

This review followed the predetermined protocol that was registered in PROSPERO (CRD42020165436). Our study was conducted in accordance with the Preferred Reporting Items for Systematic Review and Meta-Analysis (PRISMA) statement [28]. More details are available in our published protocol [29].

Our review focused on English- and Chinese-language articles that reported the brain functional effect of antidepression TMS treatment on patients with depression. All articles were searched for in the following five electronic databases from inception to 5 January 2022: PubMed, Web of Science, Embase, Wanfang Data, and China National Knowledge Infrastructure (CNKI). The key words for searching were relevant to neuroimaging and TMS treatment for depression, including depression, deprress*, transcranial magnetic stimulation, magnetic resonance imaging, and MRI. Two authors searched independently on the same day.

A study was considered for inclusion in the review if: (1) it was a peer-reviewed original case control study, cohort study or randomized controlled trial with original functional neuroimaging data that were extractable; (2) it involved resting-state or task functional neuroimaging studies or both; (3) its trial evaluated the effects of TMS as monotherapy or in combination with other treatments in patients with depression, with or without control groups; (4) it involved patients with a depressive disorder diagnosed by ICD-10 or DSM-5, whether they had unipolar depression or bipolar depression; and (5) the study reported coordinates of functional activity altered regions in either the Talairach or Montreal Neurological Institute (MNI) space and analyzed whole-brain functional activity; (6) the study used fMRI to study brain activity specific to TMS treatment for depression.

Articles were excluded if (1) they were case reports, systematic reviews, meta-analyses, letters or other secondary studies; (2) they contained subjects who overlapped and those who employed the same task; or (3) their participants had a specific subtype of depression (e.g., postpartum depression, vascular depression or poststroke depression) or a major depressive episode as a secondary diagnosis (e.g., fibromyalgia) [30].

After removing duplicates, two authors independently examined the titles and abstracts of the searched articles and excluded those that were irrelevant, and then full articles were obtained to check more details for final inclusion, using EndNote X7 software (Thomson Reuters, New York, NY, USA) throughout this process.

A customized checklist was used to assess the quality of the included articles based on related studies because no standard checklist was found for quality assessment of functional neuroimaging studies in the previous literature [31].

Two authors extracted data from the selected studies independently based on standardized pilot-tested data forms. We extracted data from the included studies based on their characteristics, the study design, the study population and details of the interventions and comparators. The details are accessible in a previously published protocol [29].

GingerALE v3.0.2 (http://www.brainmap.org) (accessed on 5 February 2022) was utilized in the present study setting conservative thresholding ALE maps at uncorrected *p* < 0.001 [32], with a minimum cluster size of 250 mm^3^. Coordinators reported in the MNI space were first changed into a Talairach space with the icbm2tal conversion tool in GingerALE [33].

In the main analysis, all included studies were pooled using ALE meta-analysis to generally explore the functional activity of brain regions that were activated by TMS. Then task-related fMRI studies and resting-state fMRI (rfMRI) studies were analyzed separately. In addition, more refined analysis was performed for activation coordinates associated with hyperactivation or hypoactivation, which were extracted from task-based studies and rest-state studies, respectively, for further exploration.

## 3. Results

### 3.1. Study Characteristics

Five articles were ultimately included after the selection process [23,34] (Figure 1). The included studies reported functional activation changes of brain regions, comprising 92 participants. The quality assessment score of all studies was above 14 points, and the average score was 15.2 (SD = 1.17) (Table 1). The TMS and fMRI details of the included studies are shown in the Appendix A).

### 3.2. ALE Meta-Analysis Results

The results are shown in Table 2. In the main analysis, we found clusters were activated associated with TMS treatment in patients with depression in the right precentral gyrus, right posterior cingulate, left inferior frontal gyrus and left middle frontal gyrus. The results are displayed in Figure 2.

With regard to resting-state studies, analysis was performed based on three publications including 22 foci [35,36,37]. Activations associated with TMS antidepressant treatment were discovered in the right precentral gyrus, right posterior cingulate, left inferior frontal gyrus and the left superior frontal gyrus, which were the same clusters that showed increased activity when pooling hyperactivation foci extracted from resting-state studies. The details are displayed in Figure 3.

An analysis examining brain areas related to TMS treatment in task state fMRI was conducted based on two studies contained three experiments, including nine foci [23,34]. Clusters in the right middle frontal gyrus, left sub-gyrus, left middle frontal gyrus and left posterior cingulate, were found to be hyperactivated. These clusters are shown in Figure 4.

No cluster of diminished activity was found for the insufficient foci in the resting-state studies or the task-related studies.

## 4. Discussion

In this paper, we systematically reviewed and performed an ALE meta-analysis on articles that reported brain functional changes related to antidepressant TMS treatment among patients with depression. To our knowledge, the present study is the first meta-analysis to evaluate patients’ brain activation associated with antidepressant TMS treatment.

In the main analysis, the resting-state studies and task-related studies were considered together. The results of the main analysis were overlapped with the results of the resting-state studies and the task-related studies. Moreover, more activated clusters in the resting-state studies were preserved in the main analysis than the task-related studies. The pooled coordinates of the main analysis showed three that were similar to clusters from the resting-state studies results, while the other cluster in the main analysis had the same coordinate as the results of the task-related studies. This may suggest that the results of the resting-state studies had more robust clusters than the task-related ones.

The results for the resting-state studies showed altered activation in the right precentral gyrus, right posterior cingulate, left inferior frontal gyrus and the left superior frontal gyrus post-treatment. Abnormal hypoactivation was found in the right precentral gyrus among patients with major depression in the resting state according to previous studies [38]. Another study showed diminished activity in the bilateral precentral gyrus was correlated with somatic depression, which was more severe than pure depression [39]. The precentral gyrus plays a role in somatic symptoms, especially pain symptoms, due to abnormal perception of pain [40]. Moreover, reduction in functional connectivity in the right precentral gyrus and regions in the right hemisphere that are associated with cognitive regulation in the default mode network (DMN) were observed [41]. In the current study, increased activation was found in the right precentral gyrus altered by TMS treatment. Few studies have reported clusters in the precentral gyrus associated with antidepressant TMS treatment. However, consistent results were shown when 1 Hz TMS was performed on healthy controls, with increased activation observed in the bilateral precentral gyrus [42]. Published studies have suggested that the right precentral gyrus plays a role in psychological or pharmacologic antidepressant treatment in patients with depression, showing hyperactivation post-treatment [43,44].

Altered activation was uncovered in the right posterior cingulate gyrus following TMS treatment in patients with depression. The posterior cingulate is part of a major node within the default mode network (DMN), a network that is responsible for the ruminative, self-referential focus of patients with depression [45]. Its regional hyperconnectivity can be related to the emotional regulation activity of DMN in major depression [46]. Additionally, as part of the limbic system, the posterior cingulate has higher functional connectivity with the lateral orbitofrontal cortex involved in non-reward in patients with depression, and also has high functional connectivity with the para-hippocampal regions related to memory [47]. These findings both suggest that the posterior cingulate, as a highly connected part, is responsible for helping the non-reward system in the lateral orbitofrontal cortex to increase effects on memory systems in depression, which contributes to rumination about sad memories and events [48]. Regional cerebral blood flow in the right posterior cingulate decreases in patients with depression, including the Brodmann area 30, the same brain area as in our results [49]. Moreover, decreased activity of the posterior cingulate has been observed in patients with bipolar disorder II depression [50]. Previous studies have demonstrated change in activity in this area after treatment for depression. Patients with depression who responded to fluoxetine showed increased activity in the right posterior cingulate [51]. Patients with treatment-resistant depression responding to deep brain stimulation also showed increased glucose metabolism and regional cerebral blood flow in the bilateral posterior cingulate [52]. We found that clusters in the right posterior cingulate gyrus were hyperactivated after TMS treatment in the present study, consistent with the antidepressant effect caused by other treatments.

Increased activation was also observed in the left inferior frontal gyrus associated with antidepression treatment. Consistent results were obtained in a previous study— clusters in the left inferior frontal gyrus exhibited increased activity after two-week TMS treatment in patients with major depression evaluated using regional cerebral blood flow [53]. Hyperactivation in the left inferior frontal gyrus was found to be significantly associated with better emotion regulation, the abnormal functioning of which is associated with the risks, course, and outcomes of major depression [54]. Previous studies found decreased activity in the left inferior frontal gyrus in patients with treatment-resistant depression, an area which is also known as a core region of the risk/action circuit related to response inhibition, partially contributing to the emotional and cognitive symptoms seen in these patients [55,56]. Consistent activity changes were detected when patients with major depression received pharmacotherapy treatment [57]. This suggests that, by affecting the left inferior frontal gyrus, TMS treatment for depression can improve the emotional or cognitive symptoms of depression.

We found that TMS treatment altered activity in the left superior frontal gyrus as well. The superior frontal gyrus plays a role in cognitive processes, such as executive function and memory retrieval [58]. Altered activity of the left superior frontal gyrus is related to executive dysfunction in patients with major depression [56]. In previous studies, decreased activation in the left superior frontal gyrus was found in patients with major depression compared with healthy controls [59,60]. In our study, increased activity was found in the left frontal gyrus post-treatment. A previous study found hyperactivation in the left superior frontal gyrus, consistent with that observed for TMS treatment, after successful treatment with an antidepressant drug [61]. It appears that TMS antidepressant treatment has the ability to normalize abnormal brain activation in patients with depression.

The results of two cognitive task-related fMRI studies indicated increased activity in the left middle frontal gyrus, the left sub-gyrus, the right middle frontal gyrus and the left posterior cingulate associated with TMS treatment for depression.

With regard to the middle frontal gyrus, it is evident that this region is responsible for excessive self-focus, negative emotion processing, and attention bias, which are associated with depression [62,63,64]. As an area involved in decision making, affective modulation and conflict resolution, diminished activation was detected in the left middle frontal gyrus in patients with major depression disorder [65,66]. In the left middle frontal gyrus, we found antidepressant-TMS-related increased activation. This region showed changes related to TMS treatment for depression. Previous studies of antidepressant TMS treatment showed altered white-matter fractional anisotropy and event-related potential in the same area [67,68]. A published review showed an increased left middle frontal gyri activation effect after TMS treatment among patients with depression by detecting P200 components [69]. Hyperactivation in the left middle frontal gyrus has also been reported in studies using other treatments for depression. Consistent activation was found in the left middle frontal gyrus in patients with major depression when undertaking a negative emotion recognition task after using selective serotonin reuptake inhibitor medication, which was associated with clinical improvement [70]. Increased activity in the left frontal gyrus was also observed in healthy controls when performing a cognition-related task [71].

Additionally, altered activation was observed in the right middle frontal gyrus. Compared to the left side of the middle frontal gyrus, the right middle frontal gyrus is considered a site of convergence of the dorsal and ventral attention networks which can interrupt the process of ongoing endogenous attention in the dorsal network and reorient attention to an exogenous stimulus. Dysfunctional connectivity in this area can lead to negative attention bias in patients with depression, which can contribute to the maintenance of depression [72,73,74]. Additionally, a crucial site of attentional control between the internal and external, right middle frontal gyrus is also a region that is genetically related to the severity of depression, mediated by changes in its spontaneous activity, indicating that the right middle frontal gyrus might have a closer relationship with depression than the left side [75]. Decreased brain activity in the right middle frontal gyrus was observed in patients with both mild cognitive impairment and depression [76,77,78]. As our study showed, TMS for depression is also related to increased activity in the right middle frontal gyrus. Several articles have reported activity changes in the right middle frontal gyrus after TMS treatment for depression. Antidepression medication increased activity in the right middle frontal gyrus among patients with major depression, consistent with TMS-induced activity changes [79].

Another brain area in which increased activity associated with TMS treatment for depression has been reported is the left posterior cingulate gyrus. Published studies have consistently reported that, when examined by positron emission tomography (PET), regional cerebral blood flow increased in the left posterior cingulate after antidepressant deep brain stimulation treatment, which suggests increased functional activation [52,80]. As with the right posterior cingulate gyrus, the left posterior cingulate gyrus is involved closely with depression within DMN [81]. The left posterior cingulate plays a significant role in arousal, as well as internal vs. external focus of thought and attentional focus, abnormal changes in which may occur in the early stage of depression [82].

Our results showed that TMS for depression was related to increase in clusters in the frontal sub-gyrus. Though only a few studies have been conducted on this region association with depression, increased activity in the sub-gyrus was observed in schizophrenia patients with prominent negative symptoms after they received treatment, and hedonic deficits were alleviated [83]. This could indicate that the frontal sub-gyrus is associated with anhedonia, which is considered a core feature of major depressive disorder, and increased activity may be related to the effects of therapy for depression [84].

Depression affects patients’ brain functional activity which can be shown by differences in neuroimaging from healthy participants. Fitzgerald conducted an ALE meta-analysis and identified hypoactivity in the bilateral middle frontal gyrus and posterior cingulate among patients with depression compared to healthy controls [85]. According to our results, antidepressant TMS treatment can increase functional activity in regions including the prefrontal gyrus and posterior cingulate gyrus, which partly overlap with areas that are associated with abnormal functional hypoactivity in patients with depression. This could indicate that TMS treatment reverses and normalizes neuronal activity in depression-related regions. Consistent patterns have been observed after treatment with selective serotonin reuptake inhibitors (SSRIs), such as paroxetine, citalopram and fluoxetine [51,86,87]. This could indicate that these patterns play a vital role in antidepressant treatment. In addition, the mechanism of TMS treatment may be similar to that for SSRI treatment, which increases the availability of serotonin by blocking serotonin uptake pumps [88]. An animal study showed a related outcome: a significantly reduced serotonin turnover rate after high-intensity rTMS, indicating a proportional increase in 5-HT concentrations [89]. However, the imaging method used, and the intervention undertaken, are inconsistent between the present study and studies that reported consistent patterns when using SSRI treatment for depression, making us hesitant to directly compare them. Hence, further clarification remains to be obtained.

Our study offers an overview of brain activity changes in patients with depression after TMS treatment. We reported specific activations that are associated with TMS for depression and discussed their possible mechanisms from the perspective of functional activity. Further research can be conducted at the cellular or molecular level or on other aspects to better explain the mechanism of TMS treatment among patients with depression. To find the optimum TMS protocol for individuals, including whether and how TMS-related parameters, such as intensity, frequency, pulse and target, differ in the effect on patients’ brain function, is worth studying.

## 5. Limitations and Future Directions

Some limitations of this study need to be considered. First, the included studies were insufficient because few related studies met the eligibility criteria due to information being unavailable or failure to report activation coordinates, resulting in a relatively small sample size. Second, although the activation change result was convergent, there was still non-negligible between-study heterogeneity in the TMS protocol. Third, neural adjunction related to TMS treatment may be presented as long-term efficacy after the completion of therapy in months or years. However, the currently included studies acquired fMRI data right after the treatment, which may not capture observational functional activity alteration post-treatment that had not occurred yet, a few days or weeks after the treatment.

We propose several future directions of research for this field. First, we need more neural imaging studies to explore the mechanisms of TMS treatment for depression. Standardization of reporting of activity coordinates in studies would contribute to better understanding of brain activity changes caused by TMS, and contribute to an effective overview of the field when summarizing related studies, enabling the identification of the neural mechanisms underpinning antidepressant TMS treatment. Follow-up studies would be a valuable addition to enable assessment of the long-term effects of TMS treatment, which may be related to cortical plasticity altering the pathological brain functional activity associated with depression.

## 6. Conclusions

Our study provides an overview of brain activity changes in patients with depression after TMS treatment. The main analysis results suggest that TMS treatment for depression can alter the activity in the right precentral gyrus, right posterior cingulate, left inferior frontal gyrus and left middle frontal gyrus. In resting-state studies, increased activation was shown in the right precentral gyrus, right posterior cingulate, left inferior frontal gyrus and left superior frontal gyrus associated with TMS treatment. In task-related studies, clusters in the right middle frontal gyrus, left sub-gyrus, left middle frontal gyrus and left posterior cingulate were hyperactivated post-treatment. We reported specific activations that are associated with TMS for depression and discussed their possible mechanisms from the perspective of functional activity and the future direction of this research field.

## Figures and Tables

**Figure 1 jpm-13-00405-f001:**
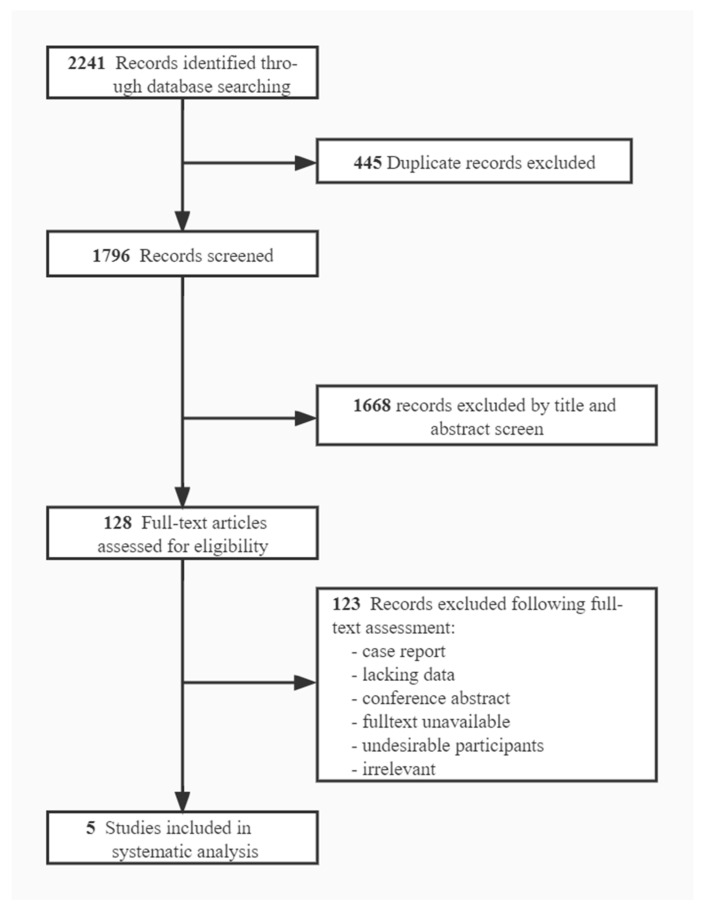
Selection process. Flow diagram of articles selected for inclusion.

**Figure 2 jpm-13-00405-f002:**
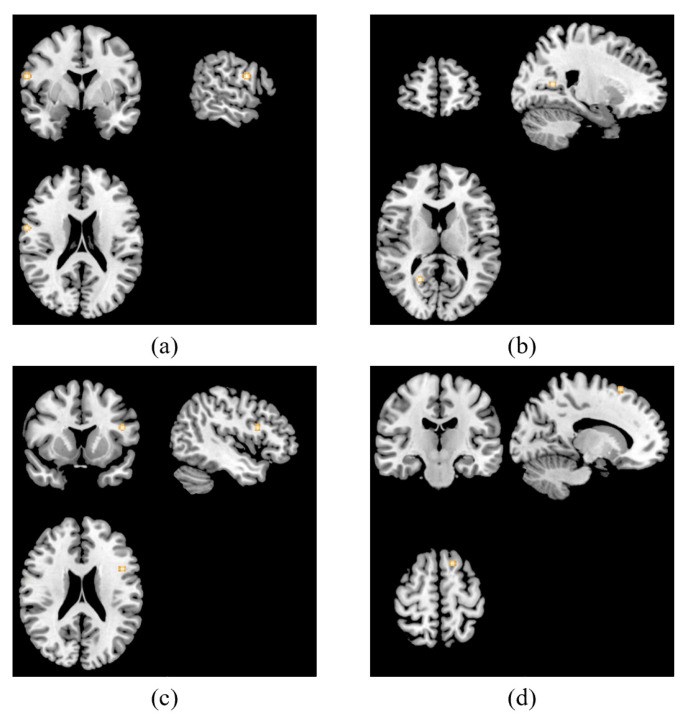
ALE results of main analysis. It illustrated that four clusters were activated post TMS treatment in the right precentral gyrus (x = 60, y = 0, z = 20) (**a**), right posterior cingulate (x = 21, y = 57, z = 12) (**b**), left inferior frontal gyrus (x = −46, y = 12, z = 22) (**c**) and left middle frontal gyrus (x = −16, y = −18, z = 62) (**d**).

**Figure 3 jpm-13-00405-f003:**
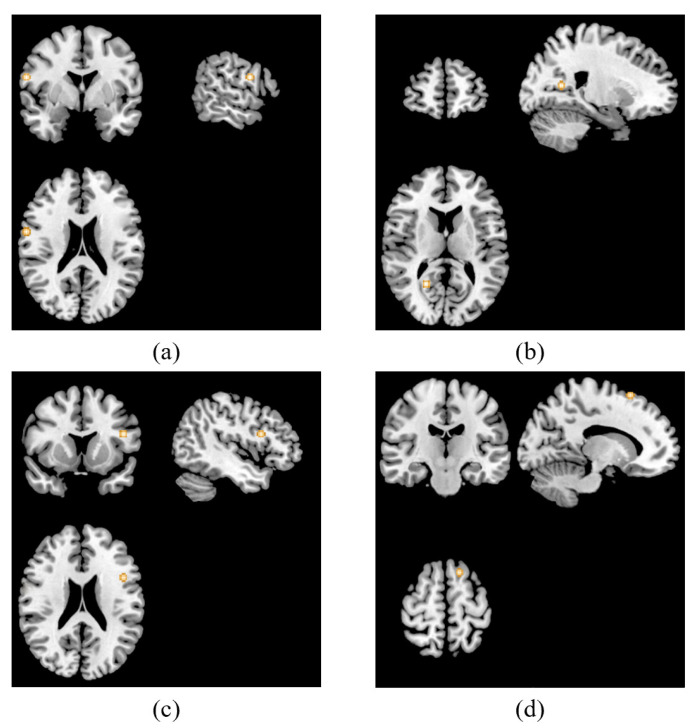
ALE results of resting-state studies. It showed that four clusters were activated associated with TMS treatment in patients with depression in the right precentral gyrus (x = 60, y = 0, z = 21) (**a**), right posterior cingulate (x = 21, y = 57, z = 12) (**b**), left inferior frontal gyrus (x = −46, y = 12, z = 22) (**c**) and left middle frontal gyrus (x = −15, y = −18, z = 63) (**d**).

**Figure 4 jpm-13-00405-f004:**
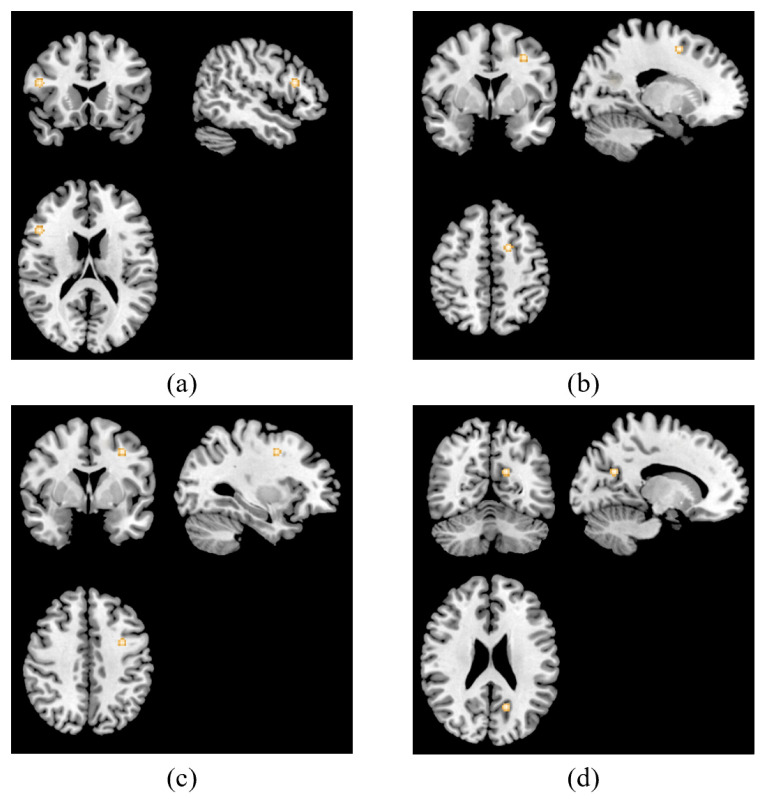
ALE results of task-related studies. Four clusters were activated associated with TMS treatment in patients with depression in the right middle frontal gyrus (x = 48, y = 20, z = 18) (**a**), left sub-gyrus (x = −18, y = 4, z = 52) (**b**), left middle frontal gyrus (x = −32, y = 42, z = 2) (**c**) and left posterior cingulate (x = −16, y = 60, z = 24) (**d**).

**Table 1 jpm-13-00405-t001:** Study characteristics.

Author (Year)	Quality Score	Severity of Depression	Sample Size (F:M)	Mean Age	Combination Therapy	Duration of Disease	Assessment Scale	Mean Score 1	Mean Score 2	Respondents:Non-Respondents	Activity Altered Region
Xingbao Li [35] (2004)	15	MDD	14(9:5)	38.2	psychiatric medication	mean: 14.1 years	HRSD	30.9	N.A.	4:10	MFG_L, HIP_L, THA_B, PUT_B, PL_B,INS_B, ORB_R, TPOmid_L, and PFC_R
Paul B. Fitzgerald [34] (2007)	14	TRD	11(5:6)	39.6	psychiatric medication	mean: 5.2 years	MADRS	33.3	25.6	6:5	MFG_B; PCUN_L
		TRD	15(8:7)	42.4	psychiatric medication	mean: 5.2 years	MADRS	34.5	24.8	9:6	PCUN_L; PreCG_L; PoSFG_L;MFG_L; IFG_R
Neacsiu AD [23] (2018)	14	MDD	5(2:3)	53.8	self-system therapy	at least 8 weeks	HRSD	19.8	3.4	5:0	ORB_R; HIP_R
Anhai Zheng [36] (2020)	17	MDD	27(19:8)	41.2	none	N.A.	HAMD	23.9	14.7	N.A.	SFG_L; LPCdor_L
Zhou Biao [37] (2022)	16	Mild to moderate depression	20(16:4)	30.2	psychiatric medication	N.A.	HAMD	18.1	4.9	N.A.	PCG_R; PoCG_R

Mean score 1 refers to mean depression-related score before TMS treatment, and mean score 2 represents mean score after TMS treatment. N.A., not available; TRD, treatment-resistant depression; F, female; M, male; MADRS, Montgomery Depression Rating Scale; HRSD, Hamilton Rating Scale for Depression; HAMD, Hamilton Depression Scale; L, left; R, right; B, bilateral; MFG, middle prefrontal cortex; HIP, hippocampus; THA, thalamus; PUT, putamen; PL, parietal lobes; INS, insula; ORB, orbitofrontal cortex; TPOmid, middle temporal cortex; PFC, prefrontal cortex; MFG, middle frontal gyrus; PCUN, precuneus; PreCG, precentral gyrus; SFG, superior frontal gyrus; PoSFG, posterior superior frontal gyrus; IFG, inferior frontal gyrus; LPCdor, dorsal lateral prefrontal cortex; PCG, posterior cingulated gyrus; PoCG, postcentral gyrus.

**Table 2 jpm-13-00405-t002:** ALE meta-analysis results.

Region	Cluster Size (mm^3^)	MNI Coordinates			ALE	*p* Value	Z Score	BA
		x	y	z	max.			
Main analysis								
Right precentral gyrus	352	60	0	20	0.0078	0.0000201	4.106458	6
Right posterior cingulate	288	21	57	12	0.0075	0.0000379	3.957385	30
Left inferior frontal gyrus	288	−46	12	22	0.0073	0.0000710	3.804717	9
Left middle frontal gyrus	288	−16	−18	62	0.0073	0.0000710	3.804717	6
Resting-state studies								
Right precentral gyrus	480	60	0	21	0.0078	0.0000137	4.194821	6
Right posterior cingulate	448	21	57	12	0.0075	0.0000299	4.013568	30
Left inferior frontal gyrus	448	−46	12	22	0.0073	0.0000611	3.842041	9
Left superior frontal gyrus	448	−15	−18	63	0.0073	0.0000610	3.842041	6
Task-related studies								
Right middle frontal gyrus	384	48	20	18	0.0063	0.0000038	4.476675	46
Left sub-gyrus	360	−18	4	52	0.0062	0.0000156	4.165172	6
Left middle frontal gyrus	352	−32	2	42	0.0060	0.0000238	4.066664	6
Left posterior cingulate	344	−16	−60	24	0.0062	0.0000156	4.165172	31

BA, Brodmann area; ALE max., ALE (activation likelihood estimation) value maximum.

## Data Availability

No additional data are available.

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
