# Peer review of "Functional Activity in the Effect of Transcranial Magnetic Stimulation Therapy for Patients with Depression: A Meta-Analysis"

_jpm, 2023, doi:10.3390/jpm13030405_

Round 1

Reviewer 1 Report

The authors propose an interesting metanalysis on the effects of rTMS on brain connectivity.

This is an important topic that both helps us to better understand the pathophysiology of depression and to elucidate the antidepressant mechanisms of rTMS.

The review is well written, but I think some points need to be clarified before it is ready for publication 

1. Line 32: premature death caused by depression is not only secondary to cognitive dysfunction but also to an increased prevalence of cardiovascular, metabolic, psychiatric and addictive comorbidities. The authors should add this with the appropriate reference.

2. Lines 48-68: Neurobiological hypotheses for the efficacy of rtMS in depression are presented in a somewhat confusing manner mixing hypotheses of cortical excitability with those of connectivity assessed by EEG or fMRI.

The authors should briefly and distinctly present the main alterations observed in depression via cortical excitability, EEG and fMRI studies and present the studies that showed that rTMS was able to correct them. 

3. Line 204-205: the sentence is not understandable and the authors have mentioned a Brodmann area without specifying it 

4. Why were other studies on the subject not taken into account, for example that of Salomon et al.https://doi.org/10.1038/npp.2013.222 or that of Liston et al. https://doi.org/10.1016/j.biopsych.2014.01.023

Author Response

Dear Reviewers:

Those comments are all valuable and very helpful for revising and improving our paper, as well as the important guiding significance to our research. We have studied the comments carefully and have made corrections which we hope meet with approval. A point-by-point response is carried out to the reviewer’s comments and suggestions.

Note:

  1. Responses are following the questions in blue;
  2. The changes are in red;

Responses to the reviewer’s comments:

  1. Line 32: premature death caused by depression is not only secondary to cognitive dysfunction but also to an increased prevalence of cardiovascular, metabolic, psychiatric and addictive comorbidities. The authors should add this with the appropriate reference.

Response:Thank you for this comment. We revised our expression on premature death caused by depression. We made following revision:

“It is a long-lasting and recurrent mental disorder that can affect people by both physical and mental dysfunction, presenting as cognitive impairment and long-term memory dysfunction, which strongly predict work disability and premature death [2-5]. Depression increased risk of premature mortality due to cardiovascular diseases metabolic, psychiatric and addictive comorbidities, causing worse prognosis [6-8].

We add this with references as follows:

“6.   Kahl K.G., Stapel B., Frieling H. Link between depression and cardiovascular diseases due to epigenomics and proteomics: Focus on energy metabolism. Prog Neuropsychopharmacol Biol Psychiatry. 2019, 89, 146-157.

  1. Leone M., Kuja-Halkola R., Leval A., D'Onofrio B.M., Larsson H., Lichtenstein P., Bergen S.E. Association of Youth De-pression with Subsequent Somatic Diseases and Premature Death. JAMA Psychiatry. 2021, 78, 302-310.
  2. Fridell M., Bäckström M., Hesse M., Krantz P., Perrin S., Nyhlén A. Prediction of psychiatric comorbidity on premature death in a cohort of patients with substance use disorders: a 42-year follow-up. BMC Psychiatry. 2019, 19, 150.”

  1. Lines 48-68: Neurobiological hypotheses for the efficacy of rtMS in depression are presented in a somewhat confusing manner mixing hypotheses of cortical excitability with those of connectivity assessed by EEG or fMRI.

The authors should briefly and distinctly present the main alterations observed in depression via cortical excitability, EEG and fMRI studies and present the studies that showed that rTMS was able to correct them.

Response:Thank you for your comment. We reconsidered the presentation of the introduction and made revision as follow:

Impaired brain plasticity can be the mechanism for depressive mood and other related symptoms, manifesting as anomalous cortical activity and dysregulation of connectivity [11]. The electroencephalogram (EEG) can be an accurate proxy of long-term clinical outcomes of patients with neuropsychiatric disorder that reflect the damage of the neural network, [12]. Patients with major depression disorder present rhythmic aberrations in prefrontal cortex recorded by EEG which may be associated with changes occurring at cortical and subcortical loops related to cognition impair-ment, showing decrease in alpha connectivity specially [13]. It not only relates to de-pression symptoms of attentional impairments, suicidal ideation, rumination, and de-pression behavioral scales score, depression severity and curative effect of depression treatment as well in resting state; but has also been proven its significance in higher cognitive abilities [13]. Brain imaging have also revealed abnormalities in brain func-tional regional brain of depression. Functional magnetic resonance imaging (fMRI) is able to detect the abnormalities of brain activity, which has shown abnormal neural activity in anterior cingulate, prefrontal cortex and orbitomeatal cortex in depression with cognition impairment [14].

  Transcranial magnetic stimulation (TMS), a noninvasive method of stimulating rele-vant brain regions beneath the coil by a magnetic field inducing electrical currents, can be a safe and effective choice for depression treatment [15,16]. TMS can alter brain region cortical activity and propagate its effect by means of exist-ing neural networks [17]. It ignites an increase in alpha connectivity in depression patients’ EGG meaning the adjusting of impaired neural functional connectivity, which present symptom improvement [18]. Repetitive TMS (rTMS), a common type of TMS treatment used clinically, can provide a long-lasting therapeutic benefit against depressive mood and depression-relevant im-pairment associated with its intrinsic ability to modulate cortical excitability and con-nectivity of the areas underlying mood and affect regulation, which is caused by plastic remodeling of synaptic connections [11,19]. Furthermore, rTMS can normalize intra-cortical facilitation in patients with major depression by performing single and paired TMS before and after treatment for evaluation, which can show increased cortical ex-citability and neurochemical transmission [20]. However, the response to TMS treat-ment is different for patients with depression and healthy subjects due to the patients’ dysfunctional neuroplasticity [21]. Thus, further understanding the neuropsychiatric effect of antidepression TMS treatment, particularly on patients with depression, is important for the optimal use of TMS. Functional neuroimaging provides unique in-sight to investigate the neuropsychiatric changes caused by TMS treatment [22].”

  1. Line 204-205: the sentence is not understandable and the authors have mentioned a Brodmann area without specifying it

Response:Thank you very much for the comments. Corrections are as follows:

“Regional cerebral blood flow in right posterior cingulate decreases among patients with depression, including Brodmann area 30, same brain area in our results [49].”

  1. Why were other studies on the subject not taken into account, for example that of Salomon et al. https://doi.org/10.1038/npp.2013.222 or that of Liston et al. https://doi.org/10.1016/j.biopsych.2014.01.023

Response:Thank you for pointing out this issue. In our present study “Functional activity in the effect of transcranial magnetic stimulation therapy for patients with depression: a meta-analysis”, we focus on the ALE meta-analysis of brain functional activity related to TMS treatment for depression. Previous studies also focus on the ALE meta-analysis functional activity, such as Fenerci et al. https://doi.org/10.1016/j.neurobiolaging.2022.06.009 or Qiu et al.  10.1016/j.neubiorev.2021.11.020. Studies mentioned in comment pay more attention to the functional connectivity response to antidepressant rTMS treatment. Therefore, these studies weren’t taken into account in our final inclusion. To clarify our inclusion, we made following correction:

“A study was considered for inclusion in the review if: 1) it was a peer-reviewed original case control study, cohort study and randomized controlled trial with original functional neuroimaging data that were extractable; 2) it involved resting-state or task functional neuroimaging studies or both; 3) its trial evaluated the effects of TMS as monotherapy or in combination with other treatments in patients with depression, with or without control groups; 4) it involved patients with a depressive disorder di-agnosed by ICD-10 or DSM-5, whether they had unipolar depression or bipolar de-pression; and 5) the study reported coordinates of functional activity altered region in either the Talairach or Montreal Neurological Institute (MNI) space and analysed whole-brain functional activity; 6) used fMRI to study brain activity specific to TMS treatment for depression.

Reviewer 2 Report

Deng and colleagues in the present review article entitled ‘Functional activity in the effect of transcranial magnetic stimulation therapy for patients with depression: a meta-analysis’, explored functional activity of brain areas, after Transcranial magnetic stimulation (TMS), in patients with depressive disorder. For this purpose, authors conducted a systematic review and a meta-analysis of the most relevant studies that evaluated  patients’ brain activation associated with antidepressant TMS treatment. Results showed altered activation in right precentral gyrus, right posterior cingulate, left inferior frontal gyrus and left superior frontal gyrus post TMS-treatment. The authors concluded the manuscript by saying that antidepressant TMS treatment can increase functional activity in region that focusing in prefrontal gyrus and posterior cingulate gyrus, which partly overlap with areas that manifested with abnormal functional hypoactivity in patients with depression. 

The main strength of this paper is that it addresses an interesting and timely question, providing an overview of brain activity changes in patients with depression after TMS treatment. In general, I think the idea of this article is really interesting and the authors’ fascinating observations on this timely topic may be of interest to the readers of Journal of Personalized Medicine. However, some comments, as well as some crucial evidence that should be included to support the author’s argumentation, needed to be addressed to improve the quality of the manuscript, its adequacy, and its readability prior to the publication in the present form.

Please consider the following comments:

A graphical abstract that will visually summarize the main findings of the manuscript is highly recommended.

Abstract: According to the Journal’s guidelines, the abstract should be a total of about 200 words maximum. Please correct the actual one.

Introduction: The ‘Introduction’ section is well-written and nicely presented, with a good balance of descriptive text and information about etiology and symptomathology that define major depressive disorder. Nevertheless, I believe that more information about pathophysiologic processes of this disease will help in introducing how abnormal connections and activity in different brain regions of the brain may serve as the pathophysiological mechanism of depression. Therefore, I suggest to begin with a theoretical explanation of depressive disorder and the role of specific brain areas, like prefrontal cortex, in the pathophysiology of this disorder. I would suggest to add more information on pathological neural substrates of depression disorder, for example focusing on ‘Dissecting Neurological and Neuropsychiatric Diseases: Neurodegeneration and Neuroprotection’ and on structural as well as functional abnormalities of prefrontal cortex that may affect patients’ cognitive impairments (https://doi.org/10.3390/biomedicines10123189). In my opinion, authors could further explore relationship between the molecular regulation of higher-order neural circuits and neuropathological alterations in this neuropsychiatric disorder (https://doi.org/10.3390/biomedicines10081897), in order to provide more insights on pathophysiological features of depression.

ALE meta-analysis results: In my opinion, this section is well organized; however, as the authors have pointed out, only 5 studies were included in the meta-analysis. Therefore, to ensure in-depth understanding and replicability of the findings, I suggest better describing in detail the few studies reported in this review, by providing a detailed description of the hypothesis, the strategies used to study TMS effects on depressive symptoms, the results and their implications. Also, in my opinion, it is necessary for the authors to present their findings using summary tables.

I would ask the authors to include a proper and defined ‘Limitations and future directions’ section before the end of the manuscript, in which authors can describe in detail and report all the technical issues that could be brought to the surface.

Tables and Figures: According to the Journal’s guidelines, please provide a short explanatory caption for the table within the text.

References: Authors should consider revising the bibliography, as there are several incorrect citations. Indeed, according to the Journal’s guidelines, they should provide the abbreviated journal name in italics, the year of publication in bold, the volume number in italics for all the references.

I hope that, after these careful revisions, the manuscript can meet the Journal’s high standards for publication. I am available for a new round of revision of this article. 

Best regards,

Reviewer
